# Investigating the Effects of a New Peptide, Derived from the *Enterolobium contortisiliquum* Proteinase Inhibitor (EcTI), on Inflammation, Remodeling, and Oxidative Stress in an Experimental Mouse Model of Asthma–Chronic Obstructive Pulmonary Disease Overlap (ACO)

**DOI:** 10.3390/ijms241914710

**Published:** 2023-09-28

**Authors:** Jéssica Anastácia Silva Barbosa, Luana Laura Sales da Silva, Juliana Morelli Lopes Gonçalves João, Elaine Cristina de Campos, Silvia Fukuzaki, Leandro do Nascimento Camargo, Tabata Maruyama dos Santos, Henrique Tibucheski dos Santos, Suellen Karoline Moreira Bezerra, Beatriz Mangueira Saraiva-Romanholo, Fernanda Degobbi Tenório Quirino dos Santos Lopes, Camila Ramalho Bonturi, Maria Luiza Vilela Oliva, Edna Aparecida Leick, Renato Fraga Righetti, Iolanda de Fátima Lopes Calvo Tibério

**Affiliations:** 1Faculdade de Medicina FMUSP, Universidade de São Paulo, São Paulo 01246-903, Brazil; jes.anastacia@gmail.com (J.A.S.B.); luana.laurasales@gmail.com (L.L.S.d.S.); julianamorellift@gmail.com (J.M.L.G.J.); fisio.elaine@hotmail.com (E.C.d.C.); sfukuzaki@gmail.com (S.F.); leandro.camargo@outlook.com.br (L.d.N.C.); tabatamaruyama@gmail.com (T.M.d.S.); tibucheski.santos.henrique@gmail.com (H.T.d.S.); suellen_karoline.m@hotmail.com (S.K.M.B.); beatriz.msaraiva@fm.usp.br (B.M.S.-R.); fernandadtqsl@gmail.com (F.D.T.Q.d.S.L.); leick51@yahoo.com.br (E.A.L.); refragar@gmail.com (R.F.R.); 2Hospital Sírio Libanês, São Paulo 01308-050, Brazil; 3Department of Medicine, University City of São Paulo, São Paulo 03071-000, Brazil; 4Departamento de Bioquímica, Universidade Federal de São Paulo (UNIFESP), São Paulo 04039-002, Brazil; camilabntr@gmail.com (C.R.B.); mlvoliva@unifesp.br (M.L.V.O.)

**Keywords:** asthma–chronic obstructive pulmonary disease overlap, protease inhibitors, inflammation, airway remodeling, oxidative stress

## Abstract

The synthesized peptide derived from *Enterolobium contortisiliquum* (pep3-EcTI) has been associated with potent anti-inflammatory and antioxidant effects, and it may be a potential new treatment for asthma–COPD overlap—ACO). Purpose: To investigate the primary sequence effects of pep3-EcTI in an experimental ACO. BALB/c mice were divided into eight groups: SAL (saline), OVA (ovalbumin), ELA (elastase), ACO (ovalbumin + elastase), ACO-pep3-EcTI (treated with inhibitor), ACO-DX (treated with dexamethasone), ACO-DX-pep3-EcTI (treated with dexamethasone and inhibitor), and SAL-pep3-EcTI (saline group treated with inhibitor). We evaluated the hyperresponsiveness to methacholine, exhaled nitric oxide, bronchoalveolar lavage fluid (BALF), mean linear intercept (Lm), inflammatory markers, tumor necrosis factor (TNF-α), interferon (IFN)), matrix metalloproteinases (MMPs), growth factor (TGF-β), collagen fibers, the oxidative stress marker inducible nitric oxide synthase (iNOS), transcription factors, and the signaling pathway NF-κB in the airways (AW) and alveolar septa (AS). Statistical analysis was conducted using one-way ANOVA and *t*-tests, significant when *p* < 0.05. ACO caused alterations in the airways and alveolar septa. Compared with SAL, ACO-pep3-EcTI reversed the changes in the percentage of resistance of the respiratory system (%Rrs), the elastance of the respiratory system (%Ers), tissue resistance (%Gtis), tissue elastance (%Htis), airway resistance (%Raw), Lm, exhaled nitric oxide (ENO), lymphocytes, IL-4, IL-5, IL-6, IL-10, IL-13, IL-17, TNF-α, INF-γ, MMP-12, transforming growth factor (TGF)-β, collagen fibers, and iNOS. ACO-DX reversed the changes in %Rrs, %Ers, %Gtis, %Htis, %Raw, total cells, eosinophils, neutrophils, lymphocytes, macrophages, IL-1β, IL-6, IL-10, IL-13, IL-17, TNF-α, INF-γ, MMP-12, TGF-β, collagen fibers, and iNOS. ACO-DX-pep3-EcTI reversed the changes, as was also observed for the pep3-EcTI and the ACO-DX-pep3-EcTI. Significance: The pep3-EcTI was revealed to be a promising strategy for the treatment of ACO, asthma, and COPD.

## 1. Introduction

Asthma, a heterogeneous disease, is a chronic inflammatory condition of the airways, resulting in variable expiratory flow limitation, which can become persistent over time [1]. Chronic obstructive pulmonary disease (COPD) is a preventable and treatable disease, mainly characterized by chronic airflow limitation and associated with an abnormal inflammatory response in the airways and lungs caused by exposure to toxic particles or gases, which contributes to a decline in lung function and disease severity [2].

The asthma–COPD overlap (ACO), a newly discovered disease entity, shows overlapping features and clinical manifestations common to both asthma and COPD [1]. The prevalence of ACO varies extensively, and no universally validated diagnostic criteria for ACO have been established, challenging clinicians and the patient population [3,4,5]. Patients with ACO show a higher frequency of exacerbation and more rapid decline in lung function and an increase in asthmatic features, such as eosinophilic airway inflammation, than those with asthma or COPD alone, indicating the predictability of inhaled corticosteroid (ICS) responsiveness in COPD [6,7,8]. For ACO treatment, GINA/GOLD suggested starting treatment for asthma, using inhaled corticosteroids (ICS) at a low or medium dose and, if necessary, adjunct treatment with a long-acting beta 2-agonist and/or a long-acting muscarinic antagonist [1]. Moreover, pulmonary rehabilitation and vaccination are advised. However, corticosteroids cause long-term side effects, such as hypertension, type 2 diabetes mellitus, osteoporosis, sleep disturbances, and gastrointestinal complications. In addition, the medication, which is usually taken daily, is costly [9].

Therefore, novel therapies, such as proteinase inhibitors of animal or vegetable origin, emerge as significant contributors to antioxidant and inflammatory conditions, employed as therapeutic agents with the potential to treat ACO [10]. The Kunitz-type protease inhibitor, a serine protease, was isolated from the seeds of *Enterolobium contortisiliquum* and called “EcTI”. It is a plant belonging to the Mimosoideae subfamily which can grow into a tree over 20 m high [11,12]. EcTI contains four cysteines with double-polypeptide-chain proteins linked by two S–S bridges [13].

EcTI has been studied in experimental tumor models of gastric cancer, and evaluated for its cytotoxic action on human fibroblasts, glioblastoma [12,14], emphysema induced by elastase, and asthma induced by ovalbumin. In both studies, the therapeutic use of EcTI reduced inflammation, tissue remodeling, and oxidative stress in the lungs of the study animals [10,15,16].

Therefore, a peptide derived from the primary sequence of EcTI, identified as pep3-EcTI, was synthesized to verify and compare the effects of the native protein (the sequence identification of pep3 is protected by patent application No. PI 0601390-2). Pep-3EcTI has eight amino acids in its composition, a molecular mass of 994.21 Da, and an isoelectric point of 9.75 [17].

The present study aimed to investigate the effects of the peptide derived from the primary sequence of pep3-EcTI, with or without corticoids, in an experimental model of ACO in comparison with models of asthma induced by ovalbumin and emphysema induced by elastase, evaluating the functional and histopathological aspects of the effects.

## 2. Results

### 2.1. Control Groups

The results for some groups regarding airway hyperresponsiveness, inflammatory markers, remodeling, and oxidative stress are shown in the Appendix A. The Pep3-EcTI showed a slight bronchodilator effect in %Ers and %Htis, but as SAL did not differ in the other parameters, we analyzed all groups using the control with the highest value. Therefore, to facilitate the visualization of the graphs and the results, we chose to show only the data for the SAL group.

### 2.2. Lung Mechanics Airway Hyperresponsiveness to Methacholine

Figure 1a shows the %Rrs of all experimental groups. The %Rrs was significantly higher in the OVA (351.69 ± 47.04), ELA (188.91 ± 63.37), and ACO (625.15 ± 81.05) groups than in the SAL (73.50 ± 8.19) group (all *p* < 0.05). In the treatment groups, ACO-pep3-EcTI (186.77 ± 39.55), ACO-DX (227.31 ± 70.49), and ACO-DX-pep3-EcTI (276.36 ± 50.56), the %Rrs was lower than that in the ACO group (all *p* < 0.05). There were no significant differences (all *p* > 0.05) between the SAL and the three treatment groups (ACO-pep3-EcTI, ACO-DX, and ACO-DX-pep3-EcTI).

Figure 1b shows the %Ers of all the experimental groups. The OVA (203.48 ± 33.79) groups showed a significantly higher %Ers than did the SAL (18.50 ± 3.02) group (*p* < 0.05). The ELA (32.49 ± 3.16) and ACO (36.4 ± 4.87) groups showed a significantly lower %Ers than did the SAL and OVA groups (all *p* < 0.05). Compared with the ACO group, the ACO-pep3-EcTI (118.91 ± 28.10) and ACO-DX (112.53 ± 19.79) groups showed no differences (all *p* < 0.05). The ACO-DX-pep3-EcTI (158.16 ± 55.73) group showed a significantly higher Ers% than did the ACO group (*p* < 0.05). The ACO-pep3-EcTI and ACO-DX groups showed no differences when compared to the SAL group (*p* > 0.05).

The %Gtis (Figure 1c) was significantly higher in the OVA (136.19 ± 24.20) and ACO (176.43 ± 21.81) groups than in the SAL (35.45 ± 6.59) group (both *p* < 0.05). The ACO group showed a significantly higher %Gtis than did the ELA (54.77 ± 6.62) group (*p* < 0.05). In the treatment groups ACO-pep3-EcTI (93.11 ± 24.82), ACO-DX (56.28 ± 17.23), and ACO-DX-pep3-EcTI (52.11 ± 8.55), the %Gtis was significantly lower than that in the ACO group (all *p* < 0.05). Compared with the SAL group, the treatment groups ACO-pep3-EcTI, ACO-DX, and ACO-DX-pep3-EcTI showed no significant differences (all *p* > 0.05).

The %Htis (Figure 1d) did not differ significantly in the OVA (72.92 ± 19.09), ELA (19.58 ± 5.43), and ACO (29.92 ± 5.76) groups compared with the SAL (46.99 ± 2.63) group (all *p* < 0.05). The ELA and ACO groups showed a significantly lower %Htis than did the OVA group (*p* < 0.05). The ACO-DX-pep3-EcTI (86.11 ± 14.47) group showed a significantly higher %Htis than did the ACO group. Compared with the SAL group, the treatment groups ACO-pep3-EcTI (56.32 ± 14.85), ACO-DX (54.08 ± 9.51), and ACO-DX-pep3-EcTI showed no significant differences (all *p* > 0.05).

Figure 1e shows the %Raw of all the experimental groups. The OVA (454.81 ± 68.68) and ACO (402.07 ± 52.21) groups showed a significantly higher %Raw than did the SAL (355.75 ± 82.47) group (all *p* < 0.05). In addition, the ACO group showed a significantly higher %Raw than did the ELA (240.27 ± 38.25) group (*p* < 0.05). The treatment groups ACO-pep3-EcTI (157.02 ± 20.02), ACO-DX (133.07 ± 59.38), and ACO-DX-pep3-EcTI (193.67 ± 45.04) showed a significantly lower %Raw than did the ACO group (all *p* < 0.05). The treatment and SAL groups showed no significant differences (all *p* > 0.05).

### 2.3. BALF Analysis

Figure 2a shows the total and differential inflammatory cell counts. The OVA and ACO groups showed significantly higher total cell counts than did the SAL (0.70 ± 0.12) group (all *p* < 0.05). The ACO (23.21 ± 2.90) group showed significantly higher total cell counts than did the OVA (10.55 ± 2.6) and ELA (6.38 ± 0.69) groups (*p* < 0.05). The total cell counts in the treatment groups ACO-pep3-EcTI (5.40 ± 1.03), ACO-DX (2.59 ± 0.25), and ACO-DX-pep3-EcTI (4.06 ± 0.89) were significantly lower than those in the ACO group (all *p* < 0.05). The treatment group ACO-DX showed no significant differences when compared to the SAL group (*p* > 0.05).

The eosinophils (Figure 2b) were significantly elevated in the OVA (4.53 ± 1.18), ELA (0.58 ± 0.20), and ACO (4.26 ± 1.61) groups compared with the SAL group (all *p* < 0.05) and were reduced in the ACO-pep3-EcTI (1.81 ± 0.31), ACO-DX (0.74 ± 0.14), and ACO-DX-pep3-EcTI (1.33 ± 0.21) groups compared with the ACO group (all *p* < 0.05). Compared with the SAL group, the ACO-pep3-EcTI and ACO-DX-pep3-EcTI groups showed attenuation (both *p* < 0.05), but the ACO group treated with dexamethasone showed no differences in the results (all *p* > 0.05).

The lymphocyte (Figure 2c) counts were significantly higher in the OVA (1.50 ± 0.38) and ACO (1.73 ± 0.40) groups than in the SAL (0.18 ± 0.03) group (both *p* < 0.05). Compared with the OVA and ELA (0.77 ± 0.38) groups, the ACO group showed elevated lymphocytes (both *p* < 0.05). The lymphocyte counts were significantly lower in the treatment groups ACO-pep3-EcTI (0.47 ± 0.18), ACO-DX (0.30 ± 0.15), and ACO-DX-pep3-EcTI 80.35 ± 0.16) compared to the ACO group (all *p* < 0.05). There were no differences among the SAL ACO-pep3-EcTI, ACO-DX, and ACO-DX-pep3-EcTI groups (*p* > 0.05).

The macrophage (Figure 2d) counts were significantly higher in the OVA (1.92 ± 0.68), ELA (3.25 ± 0.34), and ACO (7.69 ± 0.9) groups than in the SAL (0.19 ± 0.03) group (all *p* < 0.05). The ACO group showed significantly higher macrophage counts than did the OVA and ELA groups (both *p* < 0.05). The macrophage counts were significantly lower in the treatment groups than in the ACO group (all *p* < 0.05). Compared with the SAL group, the ACO-pep3-EcTI (1.71 ± 0.28) group showed attenuation (*p* > 0.05). Compared with the SAL group, the ACO-DX (0.97 ± 0.025) and ACO-DX-EcTI (0.88 ± 0.25) group showed no significant differences (*p* > 0.05).

The neutrophil (Figure 2e) counts were significantly higher in the OVA (1.8 ± 0.03), ELA (2.0 ± 0.4), and ACO (4.28 ± 0.47) groups than in the SAL (0.16 ± 0.03) group (both *p* < 0.05). Compared with the SAL group, the ACO-pep3-EcTI (1.40 ± 0.37) group showed attenuation (*p* < 0.05). Compared with the SAL group, the ACO-DX (0.57 ± 0.19) and ACO-DX-pep3-EcTI (0.88 ± 0.44) groups showed no significant differences (*p* > 0.05).

### 2.4. Lm

The Lm was significantly higher in the ELA and ACO groups than in the SAL and OVA groups (both *p* < 0.05). The ACO-treated group showed a significantly higher Lm than did the ELA group (*p* > 0.05). The Lm was significantly lower in the treatment groups ACO-pep3-EcTI, ACO-DX, and ACO-DX-pep3-EcTI than in the ACO group (all *p* > 0.05). Compared with the SAL group, the treatment groups ACO-pep3-EcTI, ACO-DX, and ACO-DX-pep3-EcTI showed no significant differences (all *p* > 0.05). The results are represented in Table 1.

### 2.5. Inflammatory Markers

Table 2 shows the levels of the inflammatory markers IL-1β, IL-4, IL-5, IL-6, IL-10, IL-13, IL-17, INF-γ, and TNF-α in the airways an alveolar septa. The airways showed significantly higher cell counts positive for IL-5, IL-6, IL-10, IL-13, IL-17, and TNF-α in the OVA, ELA, and ACO groups than in the SAL group (all *p* < 0.05) and for IL-1β and IL-4 in the OVA and ACO groups than in the SAL group (*p* < 0.05). The INF-γ count was significantly higher in the ELA and ACO groups than in the SAL group (*p* < 0.05).

The number of cells positive for IL-1β, IL-5, IL-6, IL-10, IL-13, IL-17, and INF-γ was significantly lower in the treatment groups ACO-pep3-EcTI, ACO-DX, and ACO-DX-pep3-EcTI than in the ACO group (all *p* < 0.05). There were significantly fewer IL-4-positive cells in the ACO-DX and ACO-DX-pep3-EcTI groups than in the ACO group (all *p* < 0.05), while TNF-α in the airways showed no significant difference in the ACO-pep3-EcTI, ACO-DX, or ACO-DX-pep3-EcTI group compared with the ACO group (all *p* > 0.05).

The ACO-pep3-EcTI and ACO groups showed no difference (all *p* > 0.05) in IL-4 (airways), IL-5 (alveolar walls), IL-6 (airways), IL-10 (airways and alveolar walls), IL-13 (alveolar walls), IL-17 (airways and alveolar walls), TNF-α (airways), and INF-γ (airways and alveolar walls). Compared with the SAL group, the ACO-DX group showed no difference in IL-1β (airways and alveolar walls), IL-6 (airways), IL-10 (alveolar walls), IL-13 (alveolar walls), IL-17 (airways), TNF-α (airways), and INF-γ (airways and alveolar walls; all *p* > 0.05) results.

Compared with the SAL group, the ACO-DX-pep3-EcTI group showed no difference (*p* > 0.05) in IL-5 (alveolar walls), IL-6 (airways), and IL-10 (airways and alveolar walls) results. The ACO-DX-pep3-EcTI group facilitated the reduction shown by ACO-pep3-EcTI and ACO-DX in IL-13 (airways) and IL-17 (alveolar septa).

### 2.6. Oxidative Stress Response

The absolute values of the oxidative stress markers are shown in Table 3.

The number of cells positive for iNOS was significantly higher in the OVA, ELA, and ACO groups than in the SAL group (all *p* < 0.05) in the airways and alveolar septa. In the airways, the ELA group showed significantly higher iNOS cells than did the ACO group (*p* < 0.05), while compared with the ACO group, the OVA group showed no significant differences (*p* > 0.05). In the alveolar septa, compared with the ACO group, the OVA and ELA groups showed no significant differences (*p* > 0.05).

The OVA (32.41 ± 4.65) and ACO (41.00 ± 5.15) groups showed significantly higher exhaled nitric oxide (eNO) levels than did the SAL group (12.06 ±1.52; *p* < 0.05), and the ACO group showed significantly higher ENO levels than did the ELA group (23.50 ± 3.91; *p* < 0.05). The treatment groups ACO-pep3-EcTI (16.56 ± 1.95), ACO-DX (23.10 ± 4.31), and ACO-DX-pep3-EcTI (17.04 ± 2.03) showed significantly lower ENO levels than did the ACO group (*p* < 0.05). Compared with the SAL group, the ACO-DX group showed a significant difference (*p* < 0.05). Compared with the SAL group, the ACO-pep3-EcTI and ACO-DX-pep3-EcTI groups showed no differences in the results (*p* > 0.05).

The NF-κB was significantly higher in the OVA, ELA, and ACO groups than in the SAL group (all *p* < 0.05) in the airways and alveolar septa. Its levels were significantly higher in the ELA group than in the OVA and ACO groups (all *p* < 0.05) in the airways and alveolar septa. Its levels were significantly lower in the treatment groups than in the ACO group (all *p* < 0.05) and differed significantly in the treatment groups compared with the SAL group (*p* < 0.05).

### 2.7. Extracellular Matrix Remodeling

Table 4 shows the remodeling markers MMP-9, MMP-12, TGF-β, as well as collagen fibers in the airways and alveolar septa of all experimental groups.

The MMP-9 levels were significantly higher in the OVA, ELA, and ACO groups than in the SAL group (all *p* < 0.05) in the airways and alveolar septa. The ACO group showed significantly elevated MMP-9 in the airways when compared to the OVA group, and in the alveolar septa when compared to the OVA and ELA groups (*p* < 0.05). The MMP-12 levels were significantly higher in the ACO group than in the SAL group in the airways and the alveolar septa (all *p* < 0.05) and did not differ significantly between the OVA and the SAL groups (*p* > 0.05).

Compared with the ACO group, the treatment group ACO-DX-pep3-EcTI showed no differences in the MMP-12 (airways and alveolar septa), TGF-β (airway), and collagen fibers (airways and alveolar septa; all (*p* > 0.05) results. Compared with the SAL group, the ACO-DX group showed no differences in the MMP-12 (airways and alveolar septa), TGF-β (airways and alveolar septa), and collagen fibers (airways) (all *p* > 0.05) results. Compared with the SAL group, the ACO-DX-pep3-EcTI showed no differences in the MMP-12 (airways and alveolar septa), TGF-β (airways), and collagen fibers (airways and alveolar septa) results.

### 2.8. Qualitative Analysis

Figure 3 show representative photomicrographs of the inflammatory processes, remodeling, oxidative stress, and nuclear factor in the airways. Each image was obtained at 400× magnification.

## 3. Discussion

We evaluated the effects of the pep3-EcTI peptide isolated and associated with dexamethasone in an experimental model of ACO on lung hyperresponsiveness, differential inflammatory cells, inflammatory markers, extracellular matrix (ECM) remodeling, oxidative stress markers, and mechanisms involved in the airways and alveolar septa, modifying an experimental ACO-like model [18] and treating these experimental animals with pep3-EcTI, showing that this peptide alone can partially attenuate and reverse these responses.

The ACO group showed an exacerbation in the parameters of pulmonary hyperresponsiveness to methacholine, with elevations in %Rrs, %Raw, and %Gtis compared with the baseline. The treatment groups showed a reversal, with proximal and distal bronchodilator effects.. In the work of Rodrigues et al. [10], animals exhibiting asthma and treated with the EcTI protein also showed this effect. No previous studies were performed with the pep3-EcTI or ACO model in order to elucidate the trend in results.

In the present study, compared with the OVA group, the ACO group showed a significant decrease in %Ers, and compared with the ELA group, there were no differences, which was corroborated by the decreased %Htis, maintaining the same response pattern. These results indicate the impairment of elastic recoil due to the destruction of the alveolar wall resulting from increased alveolar enlargement. This decrease in the %Ers differed from that found by Almeida-Reis et al. [19] and Theodoro-Junior et al. [16]. They found that the %Ers increased compared with that in the SAL. This difference may be attributable to the different timings of elastase instillation, which occurred at 28 days after the elastase instillation in their study, but only 7 days after instillation in our study. The lesion in the alveolar septa may not have had sufficient time to remodel itself, with greater tissue injury, and less collagen fiber content. Comparable results to those noted in our study were also found in the research of Silva et al. [20].

Alveolar damage was evidenced by the increase in Lm, which was greater in the ACO group than in the OVA and ELA groups, suggesting hyperdistension and destruction of the alveolar septa associated with the remodeling of the ECM components [21]. Changes in the Lm were reversed in all treatment groups, suggesting attenuation of the lung injury.

In the BALF fluid, the eosinophils, macrophages, and lymphocytes were significantly higher in the ACO group than in the OVA and ELA groups, which was expected, owing to an increase in the profile of the T-helped 2 cytokines [18,22]. We found a remarkable decrease in BALF positive cells in the ACO-pep3-EcTI and in the ACO-DX, indicating that the peptide and dexamethasone modulate the anti-inflammatory processes.

Also, in our study, the ACO-pep3-EcTI and the ACO-DX groups exhibited a similar response in the numbers of IL-13 positive cells in the alveolar septa. This result was also found in an ACO model treated with the peptide *Bauhinia bauhinioides* [20]. IL-13 mediates allergic responses in patients with asthma and induces bronchial hyperresponsiveness, goblet cell hyperplasia, and mucin production [23]. Consistent with our findings, in an asthma model treated with serine protease inhibitors, Lin et al. [24] showed that reduced airway hyperresponsiveness was associated with reduced IL-5, IL-6, and IL-13 in BALF fluid, and IL-4 in the serum. IL-10 and IL-13 were more significantly reduced in the ACO group than in the OVA group. In a study evaluating the plasma of patients with asthma, IL-10 and IL-13 were elevated, associating both with an important role in inflammation [25]. Elevation in the cytokines IL-5, IL-10, and IL-13 in the ACO group in the present study supports the inflammatory profile in the ACO group. Compared with the SAL group, the ACO-pep3-EcTI and ACO-Dx-pep3-EcTI groups reversed the difference in IL-10-positive cells in the airways and alveolar septa, confirming the efficacy of the treatment with pep3-EcTI.

IL-1β-positive cells are associated with aggravated COPD during exacerbations [26], as well as with severe neutrophilic asthma, in which IL-1β-positive cells were elevated in the sputum [27]. In the present study, compared with SAL, ACO increased IL-1β-positive cells in the airways and alveolar septa and reversed the ACO-DX differences, similar to the results for SAL in the airways and the alveolar septa.

T-helper cells are characterized by the production of IL-17A, IL-17F, and IL-22, which can be released by eosinophils, neutrophils, CD8^+^ T cells, basophils, and mast cells [28], also playing a decisive role in experimental models of emphysema treated with elastase [29]. In the present study, ACO-pep3-EcTI showed no differences when compared to the results for SAL and ELA. The ACO-Dx-pep3-EcTI group showed a potential reduction in IL-17 in the alveolar wall.

Huang et al. [30] correlated plasma inflammatory cytokines with lung function in patients with ACO. In the present study, TNF-α was higher in the OVA group than in the ACO group. However, the study by Kubysheva et al. [31] included humans and showed that TNF-α was elevated in the ACO group, but the increase was greater in patients with COPD. The disease severity was confirmed by correlations between the cytokine level and spirometric indicators. In Ikeda et al.’s study [18], TNF-α was elevated in the ACO-like model group. In the present study, TNF-α was elevated in both the airways and the alveolar septa in all groups when compared to the results from the SAL group, with the elevation attenuating in the treatment groups, indicating better responses in the pep3-EcTI group, which showed different responses from those of the ACO and SAL groups.

Regarding the evaluation of lung remodeling, MMP-9 degrades ECM components, and its elevation in the present study was mainly produced by inflammatory cells, such as macrophages, neutrophils, eosinophils, and fibroblasts in the airways, thus playing a key role in upregulating these proteases, resulting in degradation and collagen remodeling [29]. Compared with the SAL group, the treated groups (ACO-pep3-EcTI, ACO-DX, and ACO-DX-pep3-EcTI) showed good attenuation results for the pep3-EcTI group, as well as for the group treated with dexamethasone. The MMP-12 positive cell assessment in both the airway and the alveolar septa showed that the peptide and the peptide association with dexamethasone (ACO-DX-pep3-EcTI) reversed the structural changes. In Theodoro-Junior et al.’s model [16], the group treated with the EcTI protein showed a decrease in positive cells.

The OVA and ACO groups showed an increased fraction of ENO, a biomarker of oxidative stress associated with increased counts of macrophages, neutrophils, lymphocytes, and eosinophils. ENO is an important indicator of airway inflammation in asthma [32]. NO (nitric oxid) is crucial for the expression of the inflammatory response, collagen deposition, and the differentiation of myofibroblasts, and it can be produced by the iNOS isoform [33,34]. Compared with ELA, ACO showed increased ENO, and ACO-pep3-EcTI and ACO-DX-pep3-EcTI reversed these responses, similar to the results noted for SAL. These data suggest that treatment with pep3-EcTI modulated this response, reducing inflammatory cells. In Li et al.’s study [35], ENO was used to differentiate ACO from COPD. They found a positive correlation between FeNO and blood eosinophil count, particularly in patients with no history of inhaled corticosteroid use [35].

iNOS-derived NO plays a pro-inflammatory role by recruiting inflammatory cells, increasing collagen, and contributing to vascular remodeling, consistent with the results of the present study, as well as those obtained by Prado et al. [36]. However, when treated with the iNOS inhibitor, TGF-β was reduced in the airways and alveolar septa, due to remodeling and modulating of the productions of collagen and elastic fibers [36]. TGF-β was elevated in the airways and alveolar septa in the ACO group compared with the ELA group. Compared with the SAL group, the groups treated in the airways showed reversed responses, suggesting an important role of the peptide in this control group.

Transcription factors are a critical modulator of inflammation in severe asthma and are elevated in severe COPD [37]. The NF-κB activation is associated with inflammation, remodeling, and oxidative stress—caused by various signals, such as cytokines and pathogens—in chronic pulmonary diseases [38]. In the present study, NF-κB was significantly higher in the OVA, ELA, and ACO groups than in the SAL group, was significantly higher in the ELA group than in the ACO group, and was attenuated in the treatment groups in both the airways and the alveolar septa.

The present study has some limitations. Using the experimental model, we evaluated remodeling changes soon after the end of the experimental protocol. A test performed after a longer period of time using porcine pancreatic elastase, or an evaluation of these changes weeks or months after the end of the experiment, would have been desirable. The present study also exhibits some strengths. We used a peptide that has not been evaluated in other studies, showing the treatment potential of pep3-EcTI in the control group, with both reversed and attenuated responses regarding the behavior of ACO.

## 4. Materials and Methods

### 4.1. Experimental Design

The study protocol was approved by the Ethics Committee on the Use of Animals of the Faculty of Medicine of the University of São Paulo (project number: 1030/2018). The animals were housed in an animal facility at the Faculty of Medicine of the University of São Paulo and cared for following the Guide to the Care and Use of Laboratory Animals published by the National Institute of Health [39]. This work was developed in the Laboratory of Experimental Therapy I of the Faculty of Medicine of the University of São Paulo.

We used 84 male Balb/c mice, weighing 20–28 g, and excluded those in whom inserting the thread resulted in cannula perforation during tracheostomy and those that died prematurely, preventing the collection of behavioral and histological data. We lost a total of four animals, including one from the SAL group during the procedure, two from the ELA group before the procedure (during tracheostomy), and one from the ACO group before the procedure (during tracheostomy).

We modified the sensitization protocol developed by Toledo et al. [40] and Ikeda et al. [18]. Our protocol lasted 28 days. Figure 4 shows a schematic diagram describing the protocol. The mice were randomly divided into eight groups: (a) the SAL group (n = 7), which received inhalational and intraperitoneal injections of the saline solution; (a) the SAL-pep3-EcTI group (n = 8), which received inhalational and intraperitoneal injections of the saline solution and intraperitoneal injections of pep3-EcTI; (b) the OVA group (n = 8), which received inhalational and intraperitoneal injections of ovalbumin; (b) the ELA group (n = 6), which received intratracheal instillation of elastase; (b) the ACO group (n = 7), which received inhalational and intraperitoneal injections of ovalbumin and intratracheal instillation of elastase; (c) the ACO-pep3-EcTI group (n = 8), which received inhalational and intraperitoneal injections of ovalbumin, intratracheal instillation of elastase, and intraperitoneal injections of pep3-EcTI; (c) the ACO-DX group (n = 8), which received inhalational and intraperitoneal injections of ovalbumin, intratracheal instillation of elastase, and intraperitoneal injections of dexamethasone; and (c) the ACO-DX-pep3-EcTI group (n = 8), which received inhalational and intraperitoneal injections of ovalbumin, intratracheal instillation of elastase, and intraperitoneal injections of dexamethasone and pep3-EcTI.

### 4.2. Inhibitor Purification

The peptide was designed and synthesized based on the EcTI protein sequence, establishing the smallest structure responsible for the inhibitory function, and correlating it to the structure and specificity of the action of the protein. The peptide was called pep3-EcTI and the sequence identification is protected by patent application No. PI 0601390-2. The pep3-EcTI has eight amino acids in its composition, a molecular mass of 994.21 Da, and an isoelectric point of 9.75. It was synthesized employing a degree of purity evaluated by reverse phase chromatography in an HPLC system equal to or greater than 98%. The synthesis was carried out by an American company, WatsonBio, Houston, TX, USA.

### 4.3. Experimental Model of Asthma

The sensitization and induction of pulmonary inflammation by ovalbumin lasted 28 days, with the mice receiving a solution of 50 mg of ovalbumin (Sigma-Aldrich, St. Louis, MO, USA) emulsified with 6 mg of aluminum hydroxide-based adjuvant Alumen (Pepsamar, Sanofi-Synthelabo SA, Rio de Janeiro, Brazil) intraperitoneally (i.p.) on days 1 and 14 and challenged with 30 min of aerosol inhalation of OVA solution diluted in 0.9% NaCl (saline) on days 21, 23, 25, and 27.

### 4.4. Mice Model of Elastase-Induced Emphysema

Mice were anesthetized with inhaled isoflurane (Isofurine^®^ 1 mL/mL, Cristália LTDA, Itapira, SP, Brazil) and received intratracheal instillation of porcine pancreatic elastase (E1250-500MG, elastase from a porcine pancreas, type I, 4 IU/mg protein, 41.7 mL, 12 mg protein/mL, 5 IU/mg protein, Sigma-Aldrich, St. Louis, MO, USA) intratracheally on day 21 of the experimental protocol at a dose of 25 U EPP/100 g of body weight dissolved in 40 μL of the saline solution [14]. All animals were evaluated after 7 days of elastase administration.

### 4.5. Experimental Model of ACO

The mice in this experimental group were subjected to the two previously described experimental protocols, following the dose and date protocols [18].

### 4.6. Pep3-EcTI and Dexamethasone-Treated Groups

Injections of pep3-EcTI (2 mg/kg i.p.) were administrated 1 h before each aerosol inhalation with ovalbumin on days 22, 23, 25, and 27 of the experimental asthma protocol [8,13]. On these same days, the group treated with corticosteroids received dexamethasone (5 mg/kg i.p.).

### 4.7. Evaluation of Hyperresponsiveness to Methacholine and Exhaled Nitric Oxide Collection

On day 28, 24 h after the end of the experimental protocol, the animals were anesthetized using thiopental (50 mg/kg i.p.) and then tracheostomized, with the animals connected to a mechanical ventilation device (flexiVent, Scireq, Montreal, QC, Canada). They were ventilated with a tidal volume of 10 mL/kg, a respiratory rate of 150 cycles/min, and a sinusoidal inspiratory flow curve.

Exhaled nitric oxide (eNO) was obtained from the expiratory portion of the ventilator through an NO-impermeable balloon (Mylar Bag, Sievers, Instruments Inc., Boulder, CO, USA). After collection using the eNO balloon for 10 min, the hyperresponsiveness to methacholine was evaluated [40] by obtaining the generated pressure values and calculating the impedance airway pressure (pressure/flow) as a function of the different frequencies produced.

Measurements were obtained at the baseline after 30 s of mechanical ventilation. Subsequently, the challenge was performed with inhalation of methacholine at doses of 3, 30, and 300 mg/mL for 1, 2, and 3 min to evaluate the bronchoconstrictor response of the airways and lung parenchyma and the percentage of increase in relation to the baseline methacholine value.

The parameters—resistance of the respiratory system (%Rrs), elastance of the respiratory system (%Ers), tissue resistance (%Gtis), tissue elastance (%Htis), and airway resistance (%Raw)—were experimentally obtained using a constant phase model that could separately analyze airways and tissue [39]. At the end of the evaluation, the abdomen was opened, and intravenous blood was collected from the inferior vena cava using a heparinized syringe (1000 IU). Subsequently, the anterior chest wall was opened for the en bloc removal of the lungs and heart and fixed at a constant pressure of 20 cm H_2_O for 24 h in 4% formaldehyde for morphometric and histological analyses [41].

### 4.8. Bronchoalveolar Lavage Fluid

The procedures were performed by infusing 0.5 mL of saline three consecutive times (total volume of 1.5 mL) through the tracheal cannula using a syringe to collect the bronchoalveolar lavage (BALF). The volume recovered was centrifuged at 1000 rpm at 5 °C for 10 min, with an average recovery of 80%, followed by resuspension in 200 μL saline. Total cell counts were obtained using light microscopy with a Neubauer hemocytometer (400×). For differential counting, 100 μL of BAL was cytocentrifuged at 450 rpm for 6 min and, after drying, the slide was stained using the Diff-Quick technique. The differential cell count was determined from the finding of 300 cells/slide, and differentiation was performed following the hemacytological criteria for the differentiation of neutrophils, eosinophils, lymphocytes, and macrophages under an optical microscope with an immersion objective (1000×). At the end of the evaluation, the lungs were removed along with the heart for morphometric and histological or histochemical analyses.

### 4.9. Histochemistry and Immunohistochemistry

The lung tissues were maintained in 4 µm thick sections of paraffin, and the slides were stained as described below. For picrosirius staining (collagen fibers), the sections were dewaxed and washed in water. They were stained for 1 h in picrosirius red at room temperature and then washed in running water for 5 min. Subsequently, the sections were stained with Harris hematoxylin for 6 min and then washed in running water for 10 min.

The samples were labeled as described by Camargo et al. [22] First, deparaffinization was performed, followed by hydration, digestion, and antigenic recovery, under pressure for 1 min at 125 °C using the citrate pH 6.0 or ethylenediaminetetraacetic acid pH 9.0 buffer. Subsequently, peroxidase blockade was performed three times using 3% hydrogen peroxide at 10 volumes for 5 min, and the solution was then washed three times with phosphate-buffered saline (PBS). Subsequently, the diluted antibodies were pipetted onto the tissue, and the slides were incubated in a humidity chamber overnight (18–20 h).

After incubation in a refrigerator and washing with PBS, the slides were incubated in a humidity chamber with a secondary antibody vector for 30 min at 37 °C, washed three times for 3 min with PBS, incubated in a humidity chamber with the vector AB conjugate complex for 30 min at 37 °C, washed again with PBS, and developed in the chromogenic solution using the 3,3’-diaminobenzidine chromosome to visualize the positive cells. The sections were stained with Harris hematoxylin (Merck, Darmstadt, Germany), and the slides were mounted. The immunohistochemistry of 12 markers (IL-1β, IL-4, IL-5, IL-6, IL-10, IL-13, MMP-9, MMP-12, TNF-α, IFN-γ, nuclear factor kappa B (NF-κB), and TGF-β) was performed.

To analyze collagen fibers, the slides were stained with picrosirius red for 1 h at room temperature and washed in running water for 5 min. Subsequently, the slides were stained with Harris hematoxylin for 6 min and washed in running water for 10 min. The antibodies used for the markers and dilutions in this study are listed in Table 5.

### 4.10. Morphometric Analysis

The point-counting technique was employed using a reticle containing 50 lines and 100 points coupled to the eyepiece of a microscope (E200 MV, Nikon Corporation, Tokyo, Japan). The reticle had an area of 10^4^ µm^2^. The airways were analyzed in four fields surrounding three airways per animal. In the lung parenchyma, ten random lung fields were analyzed. All analyses were performed at 1000× magnification. The number of positive cells was counted in the reticulum area as the number of reticulum dots in the tissue area. The number of positive cells per square area of tissue was obtained, with the results were expressed as positive cells/10^4^ µm^2^ [42].

For picrosirius analysis (collagen fibers), images were captured under the Leica DM2500 microscope connected to a digital camera (Leica DFC420 Leica Microsystems, Wetzlar, Germany) using the Infinity Capture software (v 6.5.6, Lumenera Corporation, Ottawa, ON, Canada). The images were acquired using Image-Pro Plus 4.5 software (NIH, Bethesda, MD, USA), which allowed us to select from the spectrum to be developed. These shades represented the positive areas quantified in the previously determined area. We determined the volume fraction, expressed as a percentage of the total area of the frame [22].

### 4.11. Lm

The mean linear intercept (Lm) is the most-used parameter to demonstrate the presence of alveolar enlargement [17]. The reticulum with 50 lines (100 points) was coupled under an optical microscope (E200 MV, Nikon Corporation, Tokyo, Japan), and the analysis was performed by counting the intersections between the reticulum lines and the lung parenchyma. A total of 20 distinct and random fields were analyzed in the lung parenchyma under 200× magnification on slides stained with hematoxylin and eosin. Lm was calculated using the equation: Lm = 2500 μm/number of times the line and the alveolar septum intersect [22,43,44,45].

### 4.12. Data Analysis

Data are expressed as mean ± standard error and bar graphs. Statistical analyses were performed with the one-way analysis of variance, followed by the Holm–Sidak method for between-group differences. The SAL and SAL-pep3-EcTI groups were compared using the *t*-test. A *p*-value < 0.05 was considered to indicate statistical significance. All analyses were performed using SigmaPlot 11.0 software (SYSTAT Software, SPSS Inc., San Jose, CA, USA).

## 5. Conclusions

The pep3-EcTI reversed many parameters associated with hyperresponsiveness to methacholine, oxidative stress, lung injury, attenuated inflammatory response in the BALF, markers of remodeling, collagen fiber content, and NF-κB cell expression. Although more studies are required, pep3-EcTI was shown to be a promising therapeutic strategy for the treatment of ACO, as well as asthma and COPD alone, providing a solid foundation for future studies. Moreover, the experimental model of ACO developed in this study was efficient in determining the elevation in the parameters related to hyperresponsiveness to methacholine, increase in alveolar lung injury, inflammatory response, ECM remodeling, eNO expression, iNOS-positive cell count, and NF-κB-positive cell count in both the airways and alveolar walls.

## Figures and Tables

**Figure 1 ijms-24-14710-f001:**
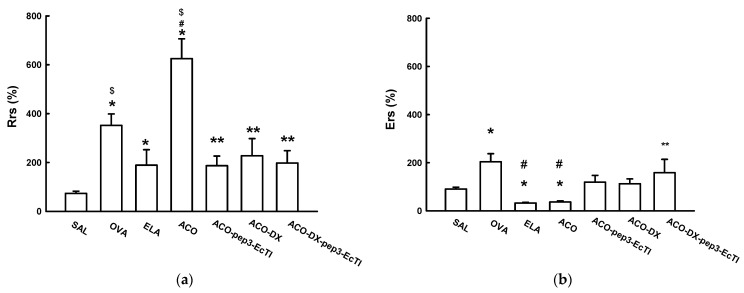
Mechanical evaluation of (**a**) %Rrs, respiratory system resistance; (**b**) %Ers, respiratory system elastance; (**c**) %Gtis, tissue resistance; (**d**) %Htis, lung tissue elastance; and (**e**) %Raw, airway resistance. * *p* < 0.05 compared to the SAL group; # *p* < 0.05 compared to OVA group; $ *p* < 0.05 compared to ELA group; ** *p* < 0.05 compared to ACO group; N = 8 for each group.

**Figure 2 ijms-24-14710-f002:**
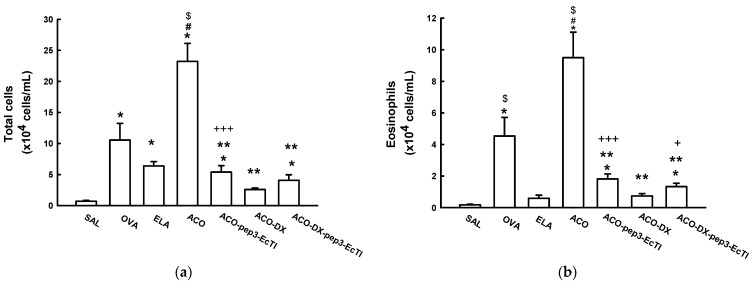
Evaluation of the number of cells in BALF. (**a**) Total numbers of cells; (**b**) eosinophils, (**c**) lymphocites, (**d**) macrophages, and (**e**) neutrophils in ×10^4^ cells/mL. * *p* < 0.05 compared to the SAL group; # *p* < 0.05 compared to the OVA group; $ *p* < 0.05 compared to the ELA group; ** *p* < 0.05 compared to the ACO group, + *p* < 0.05 compared to the ACO-pep3-ECTI group; +++ *p* < 0.05: compared to the ACO-DX-pep3-ECTI group. N = 8 for each group.

**Figure 3 ijms-24-14710-f003:**
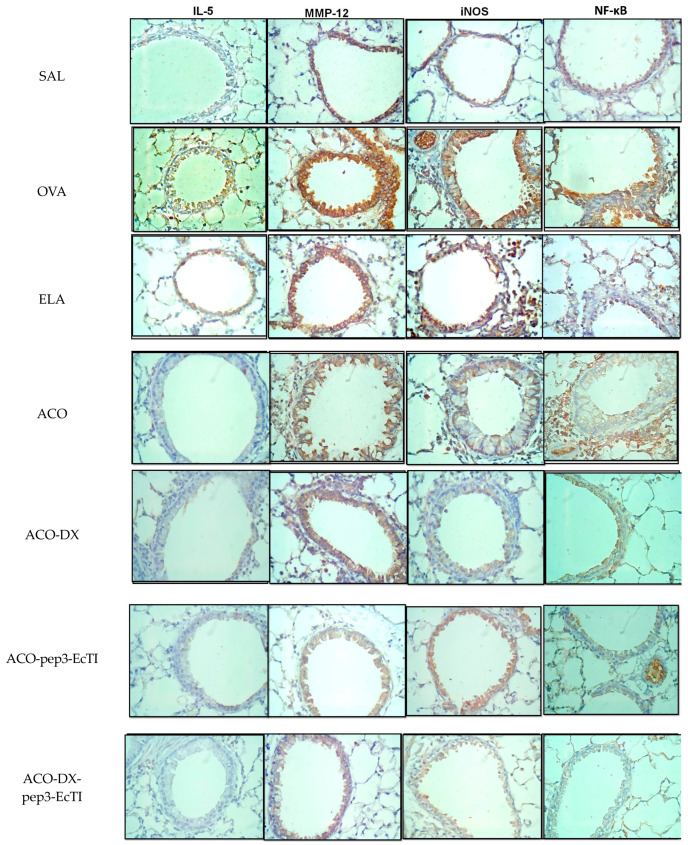
Qualitative analysis of the inflammatory marker (IL-5), the remodeling marker (MMP-12), oxidative stress (iNOS), and the signaling pathway (NF-κB). Photomicrographs of the results of the immunohistochemical analyses show the presence of inflammation in the alveolar septa (magnification of 400×). The experimental groups include SAL, OVA, ELA, ACO, ACO-pep3-EcTI, ACO-DX, and ACO-DX-pep3-EcTI.

**Figure 4 ijms-24-14710-f004:**
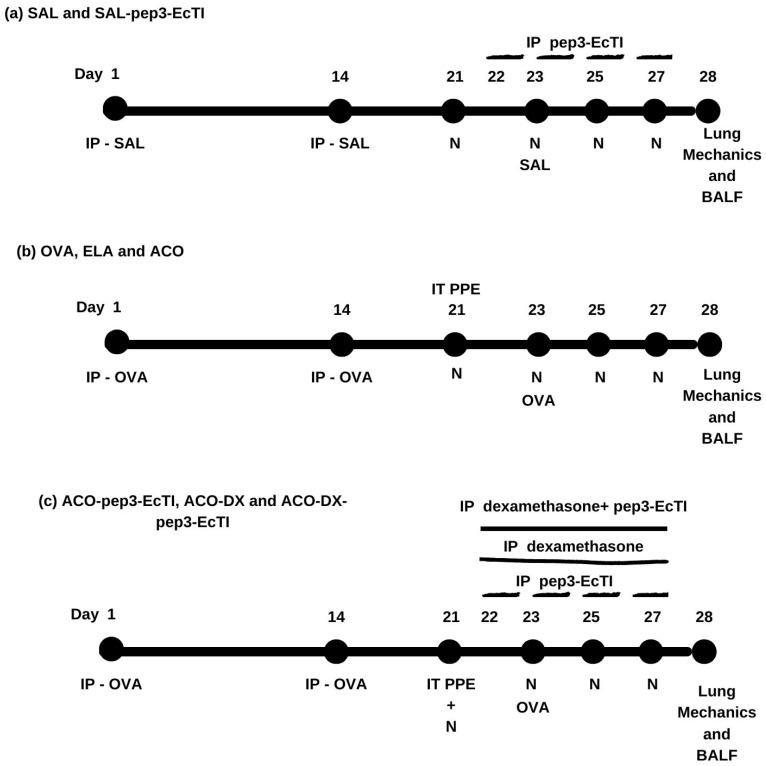
Schematic diagram describing the study protocol. (**a**) The control SAL group received intraperitoneal saline (days 1 and 14) and nebulization with saline (days 21, 23, 25, and 27). The control SAL-pep3-EcTI group received intraperitoneal saline (days 1 and 14), nebulization with saline (days 21, 23, 25, and 27), and pep3-EcTI (days 22, 23, 25, and 27). (**b**) The OVA group was sensitized with intraperitoneal ovalbumin (days 1 and 14) and received nebulization with ovalbumin (days 21, 23, 25, and 27). The ELA group received intratracheal elastase (day 21). The ACO group received intraperitoneal ovalbumin (days 1 and 14), intratracheal elastase (day 21), and nebulization with ovalbumin (days 21, 23, 25, and 27). (**c**) The treatment group ACO-pep3-EcTI received intraperitoneal ovalbumin (days 1 and 14), intratracheal elastase (day 21), nebulization with ovalbumin (days 21, 23, 25, and 27), and intraperitoneal pep3-EcTI (days 22, 23, 25, and 27). The treatment group ACO-DX received intraperitoneal ovalbumin (days 1 and 14), intratracheal elastase (day 21), nebulization with ovalbumin (days 21, 23, 25, and 27), and intraperitoneal dexamethasone (days 22, 23, 25, and 27). The treatment group ACO-DX-pep3-EcTI received intraperitoneal ovalbumin (days 1 and 14), intratracheal elastase (day 21), nebulization with ovalbumin (days 21, 23, 25, and 27), and intraperitoneal dexamethasone and pep3-EcTI (days 22, 23, 25, and 27). IP, intraperitoneal injection; IT, intratracheal instillation; N, nebulization, PPE, porcine pancreatic elastase; BALF, bronchoalveolar lavage.

**Table 1 ijms-24-14710-t001:** Absolute value of Lm.

	SAL	OVA	ELA	ACO	ACO-pep3-EcTI	ACO-DX	ACO-DX-pep3-EcTI
Linear mean intercept (µm)							
Linear mean intercept (µm)	27.27 ± 2.81	29.16 ± 1.85	42.79 ± 1.94 *	59.27 ± 1.97 *^/$^	25.81 ± 0.95 **	29.30 ± 1.31 **	27.80 ± 1.40 **

* *p* < 0.05 compared to the SAL group; ^$^ *p* < 0.05 compared to the ELA group, ** *p* < 0.05 compared to the ACO group. N = 8 for each group.

**Table 2 ijms-24-14710-t002:** Absolute values of Inflammatory markers.

	SAL	OVA	ELA	ACO	ACO-pep3-EcTI	ACO-DX	ACO-DX-pep3-EcTI
Inflammatory markers (cells/10^4^ µm^2^)							
IL-1β—Airway	0.41 ± 0.12	4.26 ± 0.44 *^/$^	1.26 ± 0.24	5.97 ± 0.56 *^/#/$^	2.91 ± 0.43 *^/^**	1.43 ± 0.30 **	2.79 ± 0.31 *^/^**
IL-1β—Alveolar septa	0.47 ± 0.17	2.63 ± 0.29 *^/$^	0.91 ± 0.22	4.06 ± 3.22 *^/$^	0.87 ± 0.17 **	0.73 ± 0.19 **	1.52 ± 0.27 ^+/++^
IL-4—Airway	1.56 ± 0.56	10.42 ± 1.09 *	3.07 ± 0.88	12.25 ± 2.30 *^/#^	8.23 ± 1.09 *	7.44 ± 1.12 *^/^**	6.35 ± 0.85 *^/^**
IL-4—Alveolar septa	1.77 ± 0.34	10.36 ± 0.59 *^/$^	4.78 ± 0.68 *^/#^	7.94 ± 0.77 *^/#/$^	4.43 ± 0.53 *^/^**	6.48 ± 0.51 *	4.66 ± 0.46 *^/^**
IL-5—Airway	1.63 ± 0.24	11.08 ± 0.76 *^/$^	6.10 ± 0.45 *^/#^	10.11 ± 0.65 *^/$/#^	3.71 ± 0.43 *^/^**	5.78 ± 0.38 *^/^**^/+/+++^	3.48 ± 0.46 *^/^**
IL-5—Alveolar septa	1.25 ± 0.19	4.41 ± 0.37 *^/$^	2.93 ± 0.64 *^/#^	8.00 ± 0.47 *^/#/$^	1.42 ± 0.36 **	2.08 ± 0.23 *^/^**^/+++^	0.82 ± 0.14 **
IL-6—Airway	1.08 ± 0.59	8.3 ± 0.54 *	5.5 ± 0.41 *^/#^	7.42 ± 0.67 *^/$^	1.67 ± 0.24 **	2.2 ± 0.35 **	2.16 ± 0.27 **
IL-6—Alveolar septa	0.48 ± 0.17	6.62 ± 0.45 *	7.20 ± 0.53 *	12.3 ± 1.34 *^/#/$^	1.86 ± 0.32 *^/^**	1.74 ± 0.32 *^/^**	1.88 ± 0.25 *^/^**
IL-10—Airway	2.13 ± 0.29	3.37 ± 0.29 *	4.06 ± 0.35 *	5.06 ± 0.40 *^/#/$^	2.31 ± 0.26 **	3.78 ± 0.29 *^/^**^/+/+++^	2.56 ± 0.24 **
IL-10—Alveolar septa	3.26 ± 0.32	4.51 ± 0.32 *	5.59 ± 0.40 *	6.6 ± 0.40 *^/#^	2.63 ± 0.50 **	3.61 ± 0.39 **	3.00 ± 0.34 **
IL-13—Airway	2.76 ± 0.34	10.4 ± 0.76 *	8.40 ± 0.78 *	11.60 ± 0.59 *^/$^	9.18 ± 0.65 *^/^**	7.63 ± 0.64 *^/^**	5.94 ± 0.45 *^/^**^/+^
IL-13—Alveolar septa	3.18 ± 0.43	9.17 ± 0.58 *^/$^	6.71 ± 0.65 *^/#^	16.9 ± 1.52 *^/#/$^	3.75 ± 0.34 **	4.22 ± 0.37 **	6.09 ± 0.44 *^/^**^/+/++^
IL-17—Airway	2.88 ± 0.23	5.16 ± 0.35 *	6.49 ± 0.30 *	7.33 ± 0.47 *^/#^	2.90 ± 0.34	4.06 ± 0.31 *^/^**	2.77 ± 0.31 **
IL-17—Alveolar septa	2.85 ± 0.31	6.49 ± 0.34 *	8.15 ± 0.48 *^/#^	8.41 ± 0.59 *^/#^	4.59 ± 0.69 **	4.92 ± 0.35 *^/^**	3.97 ± 0.66 *^/^**
INF-γ—Airway	1.02 ± 01.98	2.13 ± 0.28	3.37 ± 0.43 *	6.27 ± 0.74 *^/#/$^	1.33 ± 0.25 *^/^**^/+^	0.29 ± 0.08 **	2.23 ± 0.08 *^/^**
INF-γ—Alveolar septa	0.88 ± 0.26	2.96 ± 0.34 *	2.67 ± 0.39 *	3.19 ± 0.35 *	1.72 ± 0.42	1.12 ± 0.28 **	2.90 ± 0.20 *^/++^

* *p* < 0.05 compared to the SAL group; ^#^
*p* < 0.05 compared to the OVA group; ^$^ *p* < 0.05 compared to the ELA group; ** *p* < 0.05 compared to the ACO group; ^+^ *p* < 0.05 compared to the ACO-pep3-ECTI group; ^++^ *p* < 0.05 compared to the ACO-DX group; ^+++^ *p* <0.05: compared to the ACO-DX-pep3-ECTI group. N = 8 for each group.

**Table 3 ijms-24-14710-t003:** Absolute values of oxidative stress response.

	SAL	OVA	ELA	ACO	ACO-pep3-EcTI	ACO-DX	ACO-DX-pep3-EcTI
Oxidative stress markers							
iNOS—Airway (cells/10^4^ µm^2^)	3.34 ± 0.59	9.91 ± 1.28 *	11.79 ± 0.93 *^/^**	8.61 ± 0.48 *^/$^	2.65 ± 0.34 **	3.34 ± 0.30 **	3.43 ± 0.39 **
iNOS—Alveolar septa (cells/10^4^ µm^2^)	2.62 ± 0.38	8.36 ± 0.90 *	8.72 ± 0.62 *	9.41 ± 0.49 *	2.15 ± 0.26 **	2.85 ± 0.29 **	3.33 ± 0.30 **
eNO—Alveolar septa (ppb)	12.06 ± 1.52	32.41 ± 4.65	23.50 ± 3.91	41.00 ± 5.155	16.56 ± 5.15	23.10 ± 4.31	17.04 ± 2.03
NF-κB—Airway (cells/10^4^ µm^2^)	0.57 ± 0.19	6.74 ± 0.61 *	8.62 ± 0.77 *	6.55 ± 0.38 *^/$^	2.69 ± 0.32 *^/^**	2.28 ± 0.24 *^/^**	3.16 ± 0.32 *^/^**
NF-κB—Alveolar septa (cells/10^4^ µm^2^)	0.34 ± 0.18	6.26 ± 0.47 *	9.06 ± 0.55 *^/#^	5.91 ± 0.44 *^/$^	2.02 ± 0.25 **	2.00 ± 0.29 *^/^**	2.92 ± 0.38 *^/^**

* *p* < 0.05 compared to the SAL group; ^#^
*p* < 0.05 compared to the OVA group; ^$^ *p* < 0.05 compared to the ELA group; ** *p* < 0.05 compared to the ACO group; N = 8 for each group.

**Table 4 ijms-24-14710-t004:** Absolute values of remodeling markers.

	SAL	OVA	ELA	ACO	ACO-pep3-EcTI	ACO-DX	ACO-DX-pep3-ECTI
Remodeling markers (cells/10^4^ µm^2^)							
MMP-9—Airway	0.20 ± 0.03	3.06 ± 0.10 *^/$^	5.36 ± 0.17 *	10.27 ± 0.27 *^/#/$^	2.43 ± 0.15 *^/^**	2.09 ± 0.08 *^/^**	2.15 ± 0.11 *^/^**
MMP-9—Alveolar septa	0.08 ± 0.12	2.11 ± 0.81 *^/$^	4.43 ± 0.12 *^/#^	8.74 ± 0.31 *^/#/$^	2.63 ± 0.26 *^/^**	2.36 ± 0.14 *^/^**	1.95 ± 0.20 *^/^**
MMP-12—Airway	1.98 ± 0.37	1.97 ± 0.28	2,12 ± 0.38	3,31 ± 0.38 *^/#^	3.09 ± 0.41 ^+++^	1.02 ± 0.18 **	1.55 ± 0.30 **
MMP-12—Alveolar septa	0.66 ± 0.19	0.43 ± 0.17	2.73 ± 0.39 *^/#^	5.73 ± 0.49 *^/#/$^	5.73 ± 0.49 **	2.37 ± 0.27 **	3.24 ± 0.39 **
TGF-β—Airway	2.05 ± 0.74	5.57 ± 1.16	2.77 ± 0.40	7.06 ± 0.99 *^/$^	3.97 ± 0.50 **	3.68 ± 0.45 **	3.45 ± 0.65 **
TGF-β—Alveolar septa	0.26 ± 0.18	4.85 ± 0.30 *	2.85 ± 0.48	7.22 ± 0.99 *^/$^	2.09 ± 0.30 **	1.06 ± 0.28 **	1.88 ± 0.2 1**
Collagen Fibers—Airway	1.63 ± 0.41	12.56 ± 0.40 *^/$^	0.38 ± 0.10 *	13.76 ± 1.62 *^/#/$^	0.68 ± 0.22 **	1.38 ± 0.23 **	1.93 ± 0.3 **^/+^
Collagen Fibers—Alveolar septa	2.46 ± 0.24	11.37 ± 0.36 *^/^**^/$^	3.9 ± 0.57	9.0 ± 0.87 *^/#/$^	2.04 ± 0.16 **	5.76 ± 0.79 *^/^**^/+/+++^	2.44 ± 0.20 **

* *p* < 0.05 compared to the SAL group; ^#^
*p* < 0.05 compared to the OVA group; ^$^ *p* < 0.05 compared to the ELA group; ** *p* < 0.05 compared to the ACO group; ^+^ *p* < 0.05 compared to the ACO-pep-ECTI group;. ^+++^ *p* < 0.05 compared to the ACO-DX-pep3-ECTI group. N = 8 for each group.

**Table 5 ijms-24-14710-t005:** Markers, specifications, and dilutions.

Markers	Specifications of the Primary Antibody	Dilutions	Specifications of the Secondary Antibody	Secondary Antibody
IL-1β	SC-52012, L: A0719; Santa Cruz Biotechnology, Santa Cruz, CA, USA	1:50	L: ZG0715, Vector; Vectastain Elite ABC Kit Peroxidase (mouse IgG), Newark, CA, USA	Anti-mouse
IL-4	SC-53084, L: J1518; Santa Cruz Biotechnology, Santa Cruz, CA, USA	1:8000	L: ZF0206, Vector; Vectastain Elite ABC Kit Peroxidase (mouse IgG), Newark, CA, USA	Anti-mouse
IL-5	SC-398334, L: F1617; Santa Cruz Biotechnology, Santa Cruz, CA, USA	1:300	L: ZF0206, Vector; Vectastain Elite ABC Kit Peroxidase (mouse IgG), Newark, CA, USA	Anti-mouse
IL-6	LS-C746886, L: 144178; LSBIO, Seattle, WA, USA	1:200	L: ZF0103, Vector; Vectastin Elite ABC Kit (rabbit IgG), Newark, CA, USA	Anti-rabbit
IL-10	SC8438; Santa Cruz Biotechnology, Santa Cruz, CA, USA	1:50	L: ZF0206, Vector; Vectastain Elite ABC Kit Peroxidase (mouse IgG), Newark, CA, USA	Anti-mouse
IL-13	SC-393365, L: G1715; Santa Cruz Biotechnology, Santa Cruz, CA, USA	1:8000	L: ZF0206, Vector; Vectastain Elite ABC Kit Peroxidase (mouse IgG), Newark, CA, USA	Anti-mouse
IL-17	SC7927, L: A3113; Santa Cruz Biotechnology, Santa Cruz, CA, USA	1:100	L: ZF0103, Vector; Vectastin Elite ABC Kit (rabbit IgG), Newark, CA, USA	Anti-rabbit
TNF-α	SC-52746, L: J2418; Santa Cruz Biotechnology, Santa Cruz, CA, USA	1:5000	L: ZF0206, Vector; Vectastain Elite ABC Kit Peroxidase (mouse IgG), Newark, CA, USA	Anti-mouse
IFN-γ	SC-8308, L: B2811; Santa Cruz Biotechnology, Santa Cruz, CA, USA	1:100	L: ZF0103, Vector; Vectastin Elite ABC Kit (rabbit IgG), Newark, CA, USA	Anti-rabbit
MMP-9	SC-393859, L:6118; Santa Cruz Biotechnology, Santa Cruz, CA, USA	1:800	L: ZF0206, Vector; Vectastain Elite ABC Kit Peroxidase (mouse IgG), Newark, CA, USA	Anti-mouse
MMP-12	SC30072, L: B1910; Santa Cruz Biotechnology, Santa Cruz, CA, USA	1:400	L: ZF0103, Vector; Vectastin Elite ABC Kit (rabbit IgG), Newark, CA, USA	Anti-rabbit
TGF-β	SC-130348, L: A0219; Santa Cruz Biotechnology, Santa Cruz, CA, USA	1:700	L: ZF0206, Vector; Vectastain Elite ABC Kit Peroxidase (mouse IgG), Newark, CA, USA	Anti-mouse
iNOS	RB-9242-P, L: 9242P709; Thermo Fisher Scientific, Runcorn, UK	1:150	Thermo Fisher Scientific UK L: ZF0103, Vector; Vectastin Elite ABC Kit (rabbit IgG), Newark, CA, USA	Anti-rabbit
NF-κB	SC-8008, L: B1119; Santa Cruz Biotechnology, Santa Cruz, CA, USA	1:700	L: ZF0206, Vector; Vectastain Elite ABC Kit Peroxidase (mouse IgG), Newark, CA, USA	Anti-mouse

Abbreviations: IL, interleukin; IFN, interferon; TNF, tumor necrosis factor; MMP, matrix metalloproteinase; TGF, transforming growth factor; iNOS, inducible nitric oxide synthase; NF-κB, nuclear factor kappa B.

## Data Availability

Data are contained within the article and Appendix A.

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
