# Peer review of "Investigating the Effects of a New Peptide, Derived from the Enterolobium contortisiliquum Proteinase Inhibitor (EcTI), on Inflammation, Remodeling, and Oxidative Stress in an Experimental Mouse Model of Asthma–Chronic Obstructive Pulmonary Disease Overlap (ACO)"

_ijms, 2023, doi:10.3390/ijms241914710_

Round 1

Reviewer 1 Report

The author provides an investigation of a new peptide against inflammation during asthma-chronic obstructive pulmonary disease overlap. The idea is interesting while there are some issues listed here:

In Line 38, there is a typing error “test-t”, I assume it should be “t-test”. In line 46, it will be better to write “similar with the ACO-pep3-EcTI”, to not confuse with SAL-pep-EcTI; also, I did not find a group of “DX-EcTI”, I assume the author refers to group ACO-DX-pep-EcTI.

The Figure 1 and Figure 2 are blur, I suggest the author increase the resolution of the bar figures.

The author states that in Figure 1a, “The treatment and control groups showed no significant differences” (line107-108). I assume the “treatment groups” refer to the last three groups (ACO-pep-EcTI, ACO-DX, and ACO-DX-pep-EcTI). Based on the bars in Figure 1a, it looks there is significant difference from saline group. I suggest the author show dots in bar figure to indicate each experimental individuals.

Figure 1e has incorrect label for 5th and 6th groups. There are no “+”, “++”, or “+++” comparisons in the bar figure, thus no need to mention them in the legend.

For line 124-125, ling 135-137, ling 141, the author made statement with either “no differences” or “did not differ significantly” then followed by “all p<0.05”. This is confusing if the author did the statistical comparison correctly.

In Figure 2a and 2c, based on the size of the bars so as the SEM, it is surprise to notice that there is no significant difference between ELA and SAL groups. In line 157, statement “The treatment group ACO-DX showed the opposite result (p < 0.05)” is unclear. Not sure which result does ACO-DX group opposite to.

I suggest the author to re-write the sentence of line 161-164. Current writing confuse readers from understanding the author’s point.

Line 176-179, statement is completely opposite to the p-value.

Figure 3 has repeated labels for staining image of bottom two rows.

The author should go through their manuscript careful, analyze the data and present them well.

Reviewer 2 Report

Dear Authors,

Thank you very much for your well-written manuscript dealing with an interesting research issue. Please pay attention to the following questions and comments, pertaining to your manuscript:

1.      Line 215: Table 3 shows the markers of oxidative stress in the alveolar septa. This is Table 2 which shows the markers of inflammation in the alveolar septa. Please correct.

2.      Table 4: please maintain the same nomenclature for the term collagen fibers.

Best Regards

Round 2

Reviewer 1 Report

The authors have addressed the issue in the revised version of manuscript.

No further comments from me.